# In-fibre logic and memory via tuneable passivation–corrosion

Yuanlong Li [1,2], Weifeng Yang [1,2] ✉, Alexander V. Shokurov [1],
Manuel Reis Carneiro [1] & Carlo Menon [1] ✉

Textile electronics with digital capabilities could sense, process, and store data, while providing immersive interaction with user and their immediate surroundings. However, existing textile electronic systems are typically built on von Neumann architecture and rigid chips, limiting their seamless integration with clothing. Here, we propose a single-fibre logic/memory electronic device based on interface passivation-corrosion whose functions do not depend on traditional carrier heterojunction interfaces. The same fibre can be switched to operate as either a diode or a memristor. The diode mode remains stable under higher voltages and longer cycling periods than the state-of-the-art anion–cation heterojunction fibres. The fibre electronics are highly stretchable (up to 50%), and are compatible with industry-standard weaving techniques. We also demonstrate the application of these fibres in "AND" and "OR" logic gates, neuromorphic synapses, and textile memristor arrays. Regulated passivation-corrosion-enabled logic and memory in fibres offers a promising avenue for the next-generation textile computing.

With the advancing pace of the development of wearable technologies, electronic-textile digital systems emerge as a bleeding-edge technology in current research[1–3]. Electronic functions such as computing and data storage are now directly embedded into textiles, enabling efficient data collection[4], processing[3] and storage[5] in real time at the wearable garment-based device itself. This allows immersive interactions of the device user with their surroundings without the need to outsource processing and memory outside of the worn device itself[6–8]. However, most textile electronic systems intended for on-textile computing are typically based on von Neumann electronic architecture[4,9,10], and thus strongly depend on intrinsically rigid silicon-based integrated circuits. While extremely powerful and well-established, these rigid silicon-based circuits[3,11] (stiffness: $10^7$ to $10^9$ N·m$^{-1}$; curvature radius: $10^{-6}$ to $10^{-3}$ m) typically present a very large mechanical mismatch when interfaced with soft materials typically used in everyday clothes[12,13] (stiffness: $10^3$ to $10^5$ N·m$^{-1}$; curvature radius: $10^{-3}$ to $10^{-2}$ m). This fact brings challenges for the seamless integration of the devices into garments, wearing comfort, and daily usage, for example, washing. This aspect of the traditional von

Neumann architecture-based textile computing and data storage makes it almost impossible to completely hide the technological entity within the garment[14–16], precluding inconspicuous and unobtrusive computing in the daily apparel. To achieve true "textile computing", we must first start with the underlying architecture and create flexible computing and storage devices that match the mechanical properties and form-factor of the textile substrate itself.

In pursuit of true "textile computing", we sought to embed basic computing and memory function into a single fibre or a yarn. The fibre itself should perform the role of logic signal process and digital memory function, making it a building block for the future computing-capable electronic textiles. Currently, smart fibres capable of such functions are made via a combination of semiconducting materials brought together in one fibre/yarn to form a heterostructure interface[17–19]. This structural and electronic alignment defines the nature of charge carriers (electrons or ions) at the interface. Electronic properties of the heterojunction itself determine some key technical parameters for the operation in the device (e.g., operational voltages and polarisation speed) for use in the single-fibre textile-based

---

[1]Biomedical and Mobile Health Technology Laboratory, Department of Health Sciences and Technology, ETH Zurich, Lengghalde 5, Zurich, Switzerland.
[2]These authors contributed equally: Yuanlong Li, Weifeng Yang. ✉e-mail: weifeng.yang@hest.ethz.ch; carlo.menon@hest.ethz.ch

computing. Some commonly employed structures have their own advantages and downsides: (1) The electron/hole heterojunction route allows for excellent carrier mobility[20] and switching ratio[21]—key attributes for fast, reliable logic operations. However, these interfacial heterojunctions require complex, multi-step fabrication processes that are difficult to achieve on one-dimensional, high-curvature fibre surfaces. Traditional materials used in this process also cannot accommodate the repeated stretching and folding that textiles experience during normal use[22]. (2) The anion/cation heterojunctions[23–25] are usually based on polyelectrolyte systems, which can be applied through simple coating or fibre-spinning methods and exhibit excellent mechanical flexibility and elasticity[26,27]. On the other hand, the charge transport in these systems is unreliable under varying environment parameters, such as humidity[28] and stress[29], and stable operation is limited to low voltages[23] (usually <1 V). Perhaps most critically, neither approach supports post-fabrication reconfiguration. Once produced, these fibres are stuck in their logic and memory roles. Unlike programmable microcontrollers, they cannot switch functions or adjust operating parameters to meet evolving computing demands. To address increasingly complex edge-computing requirements in wearables, we aim to develop fibre architectures that combine the mechanical flexibility and both logic and storage capabilities, being able to fluidly interchange on demand.

In this work, we have exploited the metal/electrolyte interface passivation-corrosion effect to create a tunable in-fibre logic and memory electronics (FLAME), which does not require the design of complex heterojunction interfaces. The device consists of an intrinsically stretchable shell that encases a built-in helical aluminium yarn, which is suspended in a hydrogel electrolyte. The oxide layer on the metal surface, which bears barrier and selector functions in the complete device, can be generated and removed in situ by pre-applying voltage after the assembly of the device, which greatly reduces the manufacturing difficulty of one-dimensional fibre electronics. By adjusting the pre-applied voltage and pulse time parameters, FLAME can exhibit various electrical device morphologies, including diodes and memristors, which are critical for reconfigurable textile electronic systems. The diode-like performance in FLAME can be continuously operated for long switching cycles (~7000 cycles at ±3 V) and a high voltage range of ±8 V (but not limited to), which is much higher than the operating voltage range of the state-of-the-art anion/cation heterojunction devices. The memristor-like performance of the FLAME device enables short-term plasticity (STP) electric pulse patterns, which produce a change in response current of about 0.5 mA under a pulse stimulus with a 50 ms interval. In addition, FLAME devices exhibit excellent stretchability (up to 50%) and compatibility with modern weaving techniques. We demonstrate the application of FLAME in the stretched state for "AND" and "OR" logic gates, neuromorphic synapses, and a textile memristor array.

## Results

### Concept and design of in-fibre logic and memory

We propose to start from the basic topological unit of textiles—fibre/yarn—and incorporate in situ controllable logic and memory functions in it[30]. This bottom-up design architecture from a single yarn, not only achieves a continuous transition with the mundane yarn used as a structural matrix for the rest of the textile in terms of elastic modulus and curvature radius[22], but also allows for endogenous arrangement of logic and memory nodes during the weaving stage. This provides a dependable technical approach for textile-level computing and storage (Fig. 1a). Figure 1b shows that FLAME with a diameter of ~500 μm can pass through a standard sewing needle hole (approximately 600 μm), demonstrating its process compatibility in regard to both fineness and softness. Our FLAME consists of a three-layer structure, with the core layer being an aluminium electrode, the middle layer being a hydrogel layer of different pH values, and the sheath layer

being a stretchable silicone rubber. During normal operation, aluminium and inert conductive carbon fibres act as an electrode pair, and the hydrogel layer acts as an electrolyte layer.

Whether the FLAME behaves as a diode or memristor highly depends on the chemistry of the interface between the metal electrode and the electrolyte, which in its turn is governed by the hydrogel pH. At pH 4–7, aluminium surface passivation dominates[31–33] (Fig. 1c, Supplementary Figs. 1, 2): short-term positive voltage bias only partially induces anodic oxidation, resulting in local conversion of aluminium (Al) to alumina ($Al_2O_3$) and formation of a fuse-type memristor. A longer-term applied bias creates a continuous and dense oxide layer that prevents faradaic processes and yields a forward-cutoff diode. A pH 8–10, surface corrosion processes dominate[34] (Fig. 1d, Supplementary Figs. 1, 3): prolonged positive bias generates a loose aluminium hydroxide ($Al(OH)_3$) sheath that covers the Al fibre surface. When a reverse bias is subsequently applied, gas evolution and the locally elevated pH convert $Al(OH)_3$ into soluble $AlO_2^-$, removing the deposit and giving rise to either diode-like or memristive characteristics, depending on the voltage application time (Supplementary Notes 1, 2). A short reverse pulse only partially dissolves the $Al(OH)_3$ layer, producing memristor-like behaviour, whereas an extended reverse bias drives continuous faradaic reactions under the high local $OH^-$ concentration, restoring a diode-like response. Hence, by adjusting electrolyte pH together with the polarity and duration of the applied voltage, the same fibre can be programmed to operate as either a diode or a memristor.

Furthermore, based on the passivation and corrosion reactions on the surface of the aluminium metal electrode, we summarised the qualitative phase diagrams of FLAME's diode-like and memristor-like behaviour under acidic and alkaline conditions (Fig. 1e). Under mildly acidic conditions, low-voltage, short-time positive polarisation pretreatment leads to FLAME behaving as a memristor; high-voltage, long-time positive polarisation pretreatment will make FLAME behave as a diode. On the other hand, in mildly alkaline conditions, low-voltage, short-time negative polarisation leads to memristor behaviour again, while after high-voltage, long-time positive polarisation, FLAME starts working as a diode. The interval between the two is manifested as a series connection of a memristor and a diode. These 'phases' of operational stages of the FLAME are denoted by colours and corresponding electric circuit symbols in Fig. 1e. The hydrogel layer inside FLAME exhibits good mechanical properties under both acidic and alkaline conditions (Supplementary Figs. 4, 5), and the entire fibre can be wound and threaded, making it compatible with modern weaving techniques. To further demonstrate the practical viability of FLAME, we conducted integration trials using a commercial weaving machine (Fig. 1f, Supplementary Movie 1). FLAME was successfully treated as a functional yarn and directly incorporated into the textile structure as a warp yarn in an elastic woven textile during the automated weaving process. The woven electronic textile maintains stable performance and structural consistency under large mechanical deformations, specifically within a stretchability range of 0% to 50% (Supplementary Fig. 6).

### Mechanism of passivation-corrosion-induced in-fibre logic/ memory

For an in-depth understanding of how the passivation-corrosion mechanism of aluminium surface affects the behaviour of FLAME electronic devices, we performed a systematic analysis of pH, electrode type (Supplementary Fig. 7), electrode micromorphology, and surface elements. When the aluminium electrode is pre-treated with a positive voltage in acid, its surface aluminium metal is partially oxidised to form a dense $Al_2O_3$ layer. This oxide gradually blankets the underlying metal and slows further anodic oxidation, a process generally referred to as surface passivation. If the Al surface is only partially covered when we begin the cyclic-voltammetry (CV) scan in two

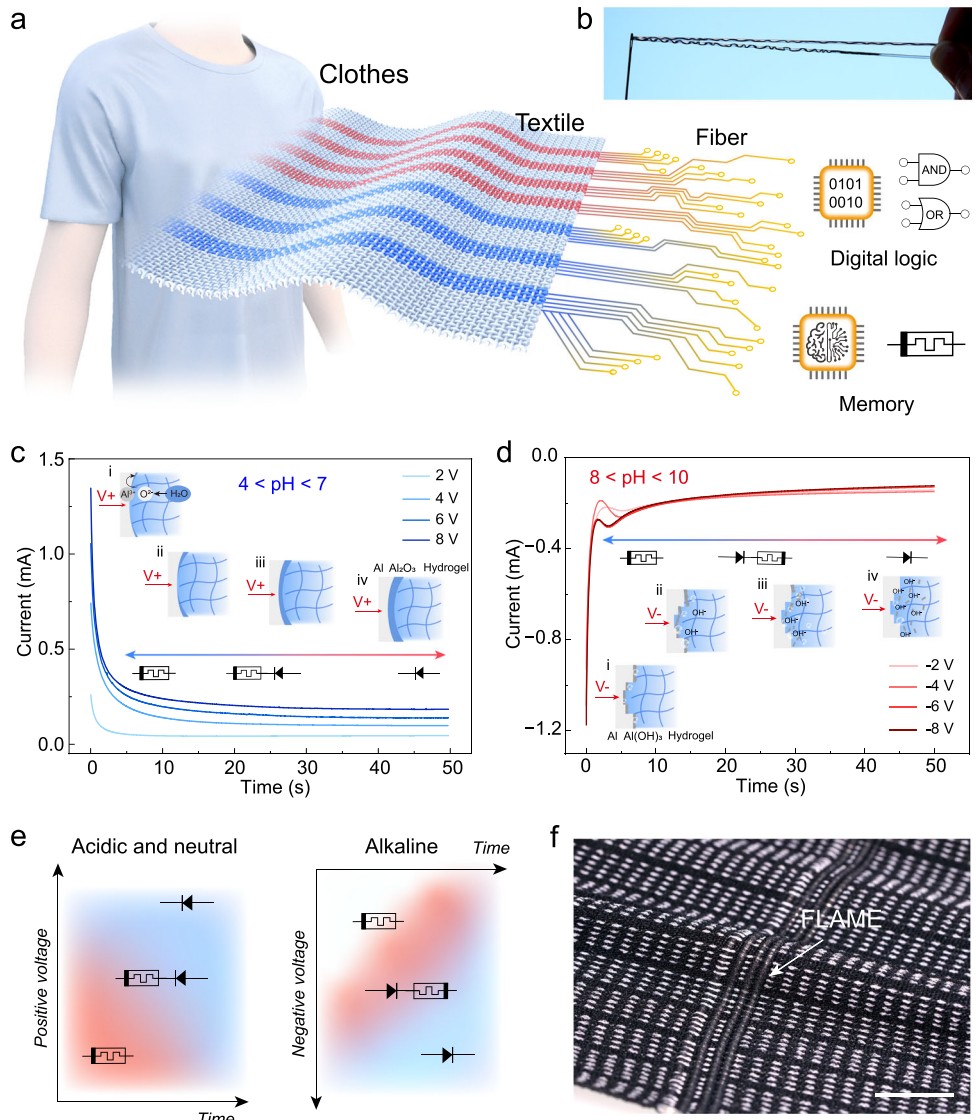

**Fig. 1 | Concept and design of in-fibre logic and memory. a** Concept of single-fibre-based logic and memory devices. **b** Digital picture of a single FLAME (20 cm length and 500 μm diameter) being threaded through a needle smoothly. **c** Pre-application of voltage for a certain time under mildly acidic conditions makes FLAME behave like a memristor or diode. The arrow underneath the graph indicates which behaviour prevails at which point in time. The insets from (i) to (iv) represent the chemical process from partial oxidation to complete oxidation of the aluminium electrode surface. **d** Pre-application of voltage for a certain amount of time

under mildly alkaline conditions makes FLAME behave like a memristor or diode. Again, the arrow underneath the graph indicates which behaviour prevails at which point in time. The insets from (i) to (iv) represent the gradual dissolution process of $Al(OH)_3$ precipitate on the surface of the aluminium electrode. **e** Qualitative phase diagrams summarising which conditions (pH, voltage, and time of application of voltage) lead to which electric element-like behaviour of the FLAME after the treatment. **f** FLAME integration trials using a commercial weaving machine. Scale bar: 0.5 cm.

electrode mode (aluminium thread being connected as a working electrode), the device behaves like a memristor: each forward sweep oxidises more and more exposed metal, enlarging the oxide area, which in turn suppresses the reaction and lowers the current (Fig. 2a, Supplementary Fig. 8a). Once the aluminium is fully coated with $Al_2O_3$, a subsequent CV scan produces almost no anodic oxidation. The oxide acts as an effective barrier, so the current–voltage curve now resembles that of a diode without a distinct forward turn-on (Fig. 2b, Supplementary Fig. 8b). Thus, the extent of oxide coverage—controlled by the pre-treatment voltage and time—dictates whether the FLAME exhibits memristive or diode-like behaviour. We can further describe the electrochemical reaction and charge transfer behaviour at the aluminium electrode interface from the perspective of energy diagram (Fig. 2c). When a positive overpotential is applied to the aluminium electrode, the position of the Fermi level of the aluminium electrode in terms of energy is lowered ($E_F \rightarrow E_{F'}$), which causes the overlap integral

for the reduced electrochemical species at the interface between the electrode and the electrolyte to be much bigger than for the oxidised species. An anodic current flows, and net oxidation occurs. As the anodic oxidation reaction proceeds, the passivation barrier on the aluminium electrode surface gradually thickens, hindering the direct contact between the electrode and the electrolyte, which gradually suppresses the Faraday reaction on the aluminium electrode surface. We performed a series of SEM characterisations that follow the oxide film formation on the aluminium electrode as the applied positive bias increases in mildly acidic conditions (Fig. 2d, Supplementary Figs. 9, 10). The process of the stripe-shaped oxide layer on the surface of the aluminium electrode gradually growing and completely covering its surface further confirms our hypothesis.

Under mildly alkaline conditions, a positive bias oxidises the aluminium surface, leading to the formation of a porous $Al(OH)_3$ layer. During the forward (anodic) sweep of a cyclic voltammogram, this

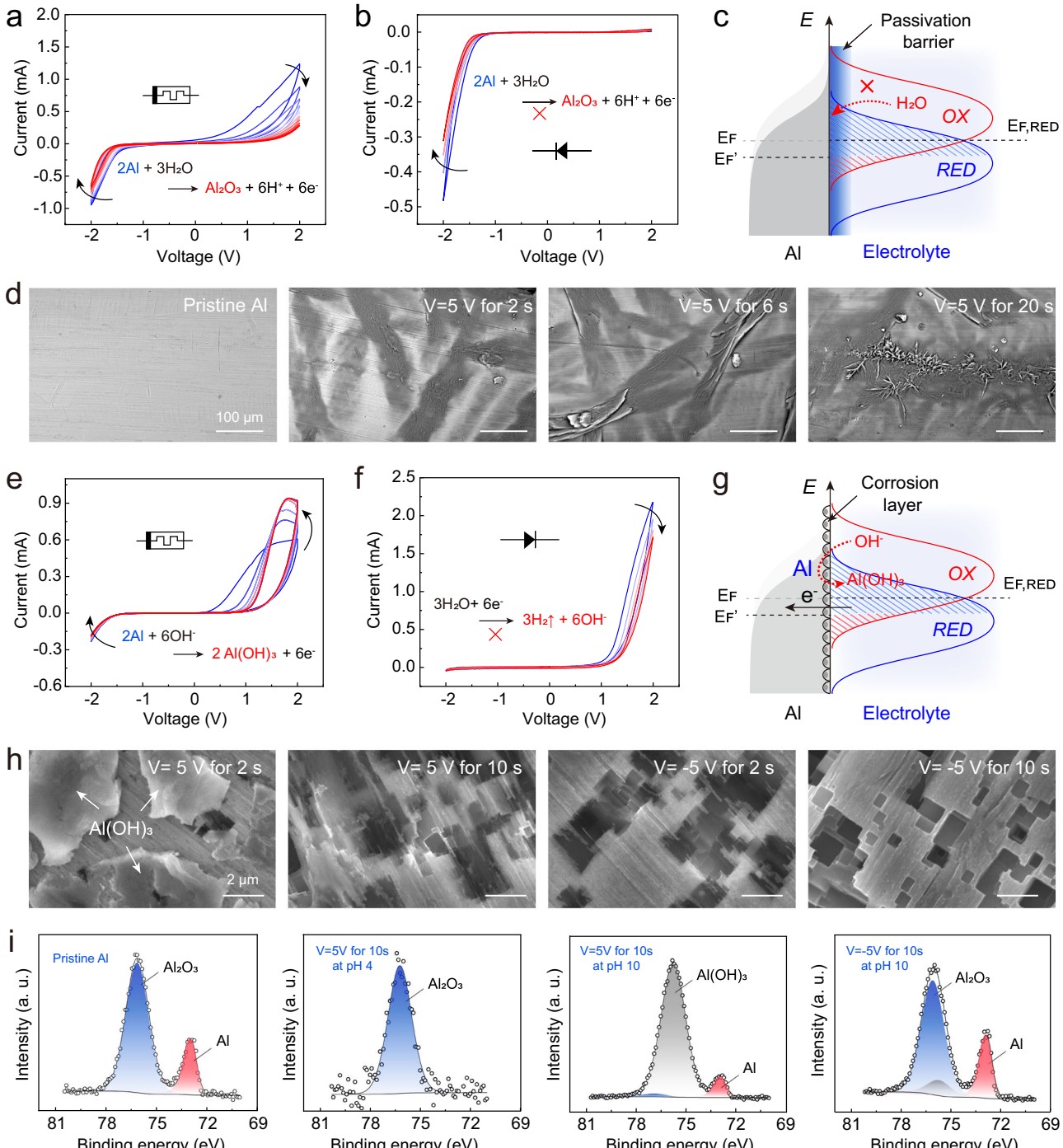

**Fig. 2 | Mechanism of passivation-corrosion-induced switching between in-fibre logic and memory functions. a** IV curve of the passivation-induced memristor-like device under acidic conditions. **b** IV curve of the passivation-induced diode-like behaving device under acidic conditions. **c** Schematic of the energy diagram of the passivation barrier blocking charge carrier transport in the interfacial Faraday reaction process. **d** SEM micrographs of the passivation process on the aluminium electrode surface, showing the gradual formation of an aluminium oxide layer. **e** IV curve of the corrosion-induced memristor-like device under alkaline conditions. **f** IV curve of corrosion-induced diode-like behaviour under alkaline conditions. **g** Schematic of energy diagram of corrosion layer promoting charge carrier transport in the interfacial Faraday reaction process. **h** SEM micrographs of the aluminium hydroxide layer formed during the corrosion process on the aluminium electrode surface and elimination of Al(OH)₃ by application of reverse voltage. **i** Al 2$p$ X-ray photoelectron spectroscopy (XPS) spectra for the surface of aluminium electrode, recorded at different device states: (a) pristine aluminium surface, (b) after pre-treatment in acidic conditions, (c) after pretreatment in alkaline conditions, and (d) after application of reverse voltage in alkaline conditions.

loose film is breached, exposing fresh metal; each new site supports further Faradaic reactions, so the anodic current continues to rise. The behaviour of the system in the opposite sweep direction is more complex. Quite likely, in the reverse sweep, water reduction raises the local OH⁻ concentration, and the excess hydroxide helps dissolve the Al(OH)₃ layer, momentarily liberating the aluminium surface, which is reflected by a small peak on the curve. Continuing application of voltage though, leads to further suppression of the water electrolysis, leading to a progressive fall in the cathodic current (Fig. 2e, Supplementary Fig. 11a). If a reverse voltage is continuously applied to the

aluminium electrode after aluminium hydroxide is formed, a large amount of hydroxide ions will be formed in the aluminium electrode region due to the electrolysis of water. Therefore, if a CV sweep is performed, the corrosion reaction of aluminium at a positive potential will be promoted, while the electrolysis reaction of aluminium at a negative potential will be suppressed, resulting in a diode-like IV curve (Fig. 2f, Supplementary Fig. 11b). From the perspective of energy diagram (Fig. 2g), a positive over-potential lowers the aluminium electrode's Fermi level, increasing its electronic overlap with reducible species in the electrolyte. This expands the electron overlap between the metal state and the reducible species in the electrolyte, thereby promoting electron transfer (redox) and corroding the aluminium metal. As oxidation proceeds, more new aluminium-solution interfaces are exposed, adding additional reactive sites and further accelerating the Faradaic process. Figure 2h, Supplementary Figs. 12, 13 show a series of SEM images showing the formation of loose bulk $Al(OH)_3$ under applied forward bias in a weak alkaline solution and the elimination of $Al(OH)_3$ upon application of reverse voltage, which further supports our hypothesis.

We further validated our findings by characterising the chemical bonds on the aluminium surface at different reaction stages using X-ray photoelectron spectroscopy (XPS) (Fig. 2l, Supplementary Fig. 14). XPS is an excellent tool to distinguish the nature of the chemical bonds on the surface and to determine precise aluminium oxidation states by their binding energies (Al 2$p$ region). Pristine aluminium shows two peaks: 76.3 eV for the native $Al_2O_3$ film and 72.75 eV for the underlying metallic Al[31]. After anodic passivation in acid, a compact oxide blankets the surface; only the 76.3 eV peak remains, indicating complete coverage of the metal (at least within the depth reach of XPS capabilities). In mild alkali, corrosion converts the surface to $Al(OH)_3$, as evidenced by a new peak at 75.75 eV, while the $Al_2O_3$ signal all but disappears. Applying a reverse bias in the same alkaline medium locally raises the hydroxide concentration, dissolving $Al(OH)_3$. As the hydroxide layer thins, its 75.75 eV peak diminishes, and the characteristic $Al_2O_3$ and metallic Al peaks re-emerge, confirming partial restoration of the original surface states.

## Fibre-based diode for rectification and logic gate

Because of the electrical behaviour of the aluminium electrode under different pH environments and different pretreatment voltages, it can play a similar function to the diode in electronic components. These properties can be exploited to construct a fibre-like diode that would show similar effects. As shown in Fig. 3a, we design a stretchable fibre-like diode which consists of an intrinsically stretchable shell and a built-in helical aluminium yarn, which shows a reversible strain of 50% (Fig. 3b, Supplementary Fig. 15). Hydrogels with different pH are used as filling of the tube and act as electrolyte surrounding the aluminium yarn. Aluminium yarn is left protruding on one end of the yarn to form one electrical contact of the diode-fibre. On the other end of the diode fibre, an inert carbon electrode is placed at the opening of the silicon tube, closing the shell and providing the second electrical contact for the yarn diode. In mildly acid (pH = 4), applying a positive voltage to aluminium (and thus a negative voltage to the carbon electrode) forms a passivation layer that prevents anodic oxidation from occurring, resulting in low current (Fig. 3c). When a negative voltage is applied, some electrolysis of water still occurs on the aluminium electrode surface, resulting higher reverse current forward current (Supplementary Movie 2). In mildly alkaline (pH = 10), pretreatment with a positive voltage corrodes the aluminium and forms $Al(OH)_3$ flakes. This rough surface can be better oxidised under forward bias, resulting in higher current. However, under reverse bias, excess hydroxide ion inhibits reduction, resulting in lower current (Fig. 3d). Such electrochemical behaviour of the device ultimately results in the rectification effect, such as in a diode (Supplementary Table 1).

The fibre-based diode with different pH values were subjected to electrical rectification tests with different amplitude voltages, as shown in Fig. 3e. By applying a square wave voltage function with an amplitude from ±1.5 V to ±8 V at pH = 4, the output of the device showed good rectification performance. Importantly, the fibre-based diode is capable of switching rectification direction in a controlled fashion, with different rectification directions achieved by simply changing the pH value of the hydrogel portion with the same electrode pair. In the case of pH = 4, the diode exhibits conduction from the carbon electrode to the aluminium electrode. In addition, we also tested the stability of the diode-like device by conducting a rectification test at ±3 V for 24 h and about 7000 cycles (Fig. 3f). The results showed that the rectification voltage of the device did not drop significantly, showing good stability.

Apart from AC rectification, diodes can be used in more advanced electric circuits utilising this function. Here, we developed an "all-fibre" stretchable logic gate (Supplementary Fig. 16), including "OR" and "AND" gates, which constitute the basic logic unit of textile electronic systems. By sewing 2 diode-like fibre devices in parallel, the OR and AND gates can be implemented in different connections, as shown in Fig. 3g, h, respectively. On the textile, carbon black-based stretchable resistors are also connected by series/parallel connections to complete the corresponding logic gate configurations. In the constructed flexible OR gate, 2 fibre-like diode devices are connected in parallel and then connected in series with a stretchable resistor, and by applying either a low or a high voltage level at the other end of the two diode-like devices, the final output signal (either high or low) can be controlled. When a high level is applied to either the $V_{in}$ a or $V_{in}$ b side, a high-level signal can be detected at $V_{out}$ even at 50% strain (Fig. 3i, j). For the AND gate, by connecting two diodes in parallel with two resistors as shown in Fig. 3k, different level responses can be measured by applying different levels at the two class diodes (Fig. 3l). While the unloaded (none or either input is activated) signal is non-zero, the evidently much different and higher-level response is recorded only when both inputs are activated, clearly showing the textile device functioning of the AND gate. To assess the reliability of actual wearability, we conducted comprehensive durability and biocompatibility tests using the OR logic gate textile as an example. Experimental results show that the device exhibits excellent environmental stability: the logic output remained stable under temperature fluctuations from 32.5 °C to 60.1 °C (Supplementary Fig. 17a, b) and humidity variations from 35.42% to 78.43% RH (Supplementary Fig. 17c, d); even after 10 standard water washes, its electrical performance showed no significant degradation (Supplementary Fig. 18).

## Textile-based memristor device

By utilising the phenomenon of the generation of oxide or hydroxide on the surface of metallic aluminium, the studied system can exhibit memristor-like electrical properties and can thus be used in the realisation of various neuromorphic functions. Inspired by biological ion channels of neurotransmitters in synapses[35,36] (Fig. 4a), we designed and fabricated a textile-based memristor device. FLAME can exhibit different memristor behaviours under acidic (Supplementary Figs. 19, 20, Supplementary Movie 3) and alkaline conditions (Supplementary Movie 4). We focus on the memristor behaviour under alkaline conditions in the main text. The basic principle is to pre-treat the metal aluminium through oxidation reaction in an alkaline environment (Fig. 4b). The metallic aluminium undergoes a redox reaction, losing electrons, generating $Al(OH)_3$, i.e., corrosion on the surface of the metallic aluminium is induced. Then, a negative voltage is applied to the aluminium metal, inducing an electrochemical reduction at the electrode. However, during the pretreatment process, a large amount of hydroxide ions is generated on the surface, the local pH value increases, and part of the $Al(OH)_3$ will be converted into aluminate ions. The reduction reaction will also lead to the evolution of gas,

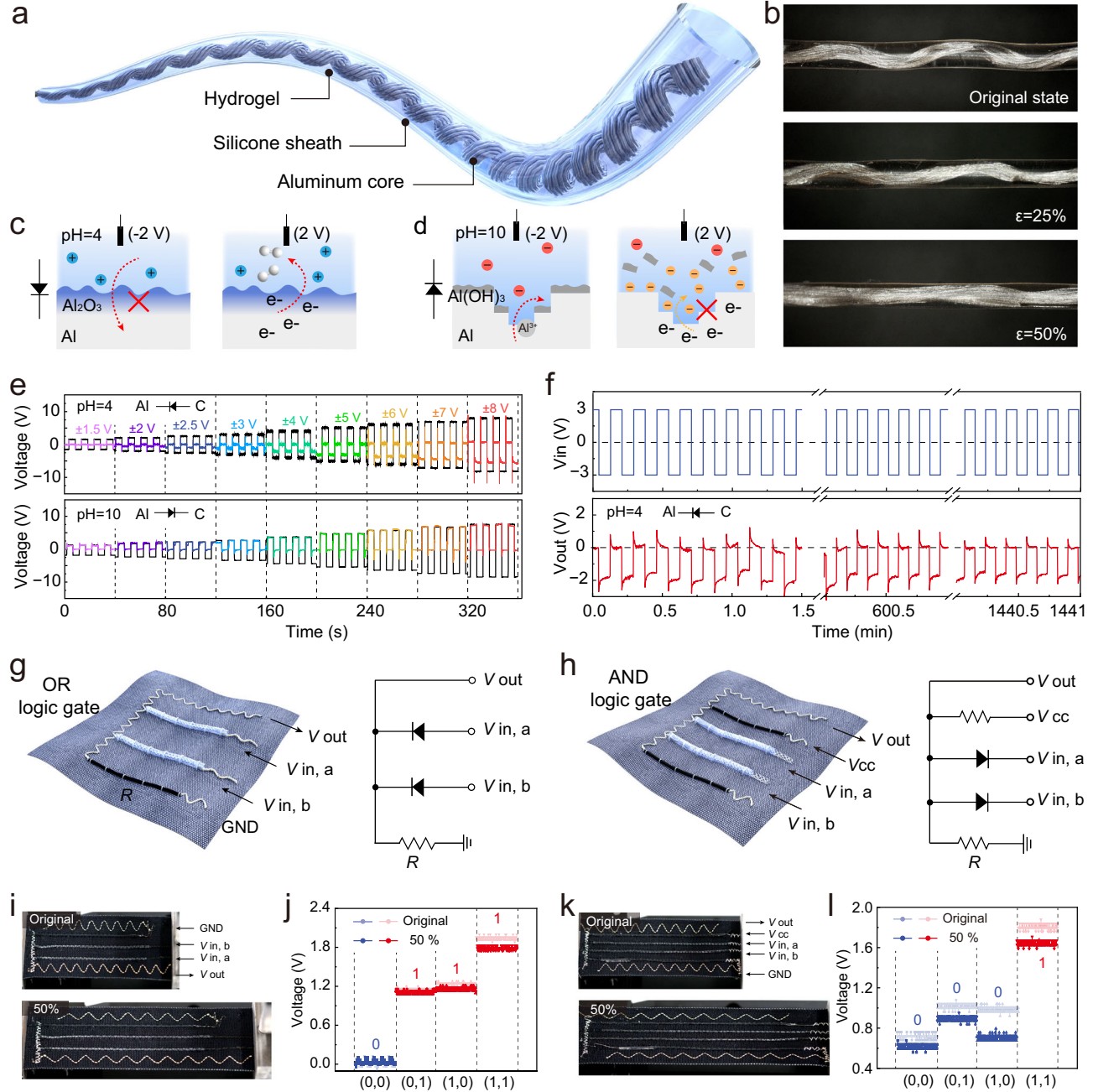

**Fig. 3 | Fibre-based diode for rectification and logic gate. a** Schematic of a fibre-based logic device. Schematic diagram of (electro)chemical processes underpinning the logic device operation in **b** acidic and **c** alkaline environments. **d** Photographs of fibre logic devices under mechanical tensile stress. Red anions: Hydroxide in mildly basic electrolytes. Yellow anions: Hydroxyl ions produced by electrolysis of water on the surface of the aluminium electrode. **e** Voltage rectification of applied voltage square waves of different amplitudes by the devices operating under acidic and alkaline conditions. **f** The results of long-time rectification tests, showing performance of a freshly made device in the minutes, hours, and day timeframes. **g** Schematic of an OR gate constructed based on a fibre logic device and other flexible electric circuit elements. **h** Schematic of an AND gate constructed based on a fibre device and other flexible electric circuit elements. **i** Photograph of an OR gate constructed based on a fibre logic device in its initial state and stretched to 50% strain. **j** Electrical output of a textile OR logic gate as a function of its inputs. **k** Photograph of an AND gate constructed based on a fibre logic device in its initial state and stretched to 50% strain. **l** Electrical output of an AND logic gate as a function of its inputs.

which will break away the $Al(OH)_3$ coating flakes on the surface. It can be seen from the SEM image in Fig. 4b that after negative polarisation of the aluminium metal, the amount of $Al(OH)_3$ floccules on the surface is significantly reduced. Furthermore, under alkaline conditions, the performance of this memristor does not significantly degrade during 250 cycles (Supplementary Fig. 21).

As a proof of concept, we show that such electrochemical behaviour can be used to mimic some of the basic functions found in biological synapses. We demonstrated that the conductivity of the memristor fibre can be increased or decreased by a continuous voltage sweep of a given polarity (Fig. 4c). The output current of the device changes depending on the input voltage sign and time of application, meaning that the overall impedance of the system can be controllably decreased or increased in this fashion. To mimic the short-term plasticity (STP) of the synapse, we applied multiple voltage pulses to the textile-based memristor and recorded current spikes in accordance

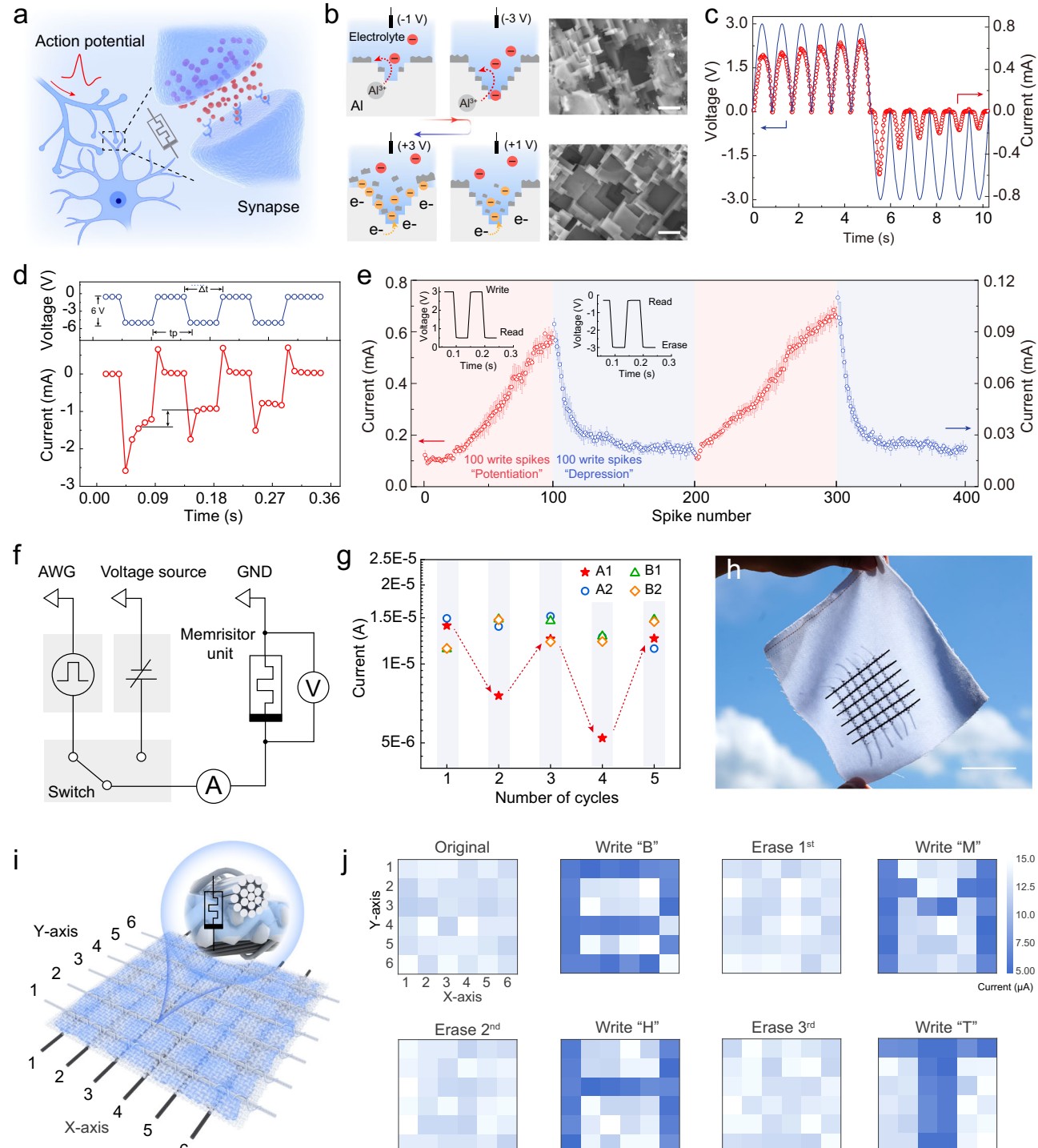

**Fig. 4 | Textile-based memristor device. a** Schematic diagram of neuronal signalling. **b** Schematic illustration of the operation of a fibre memristor in an alkaline environment. Red anions: Hydroxide in mildly base electrolytes. Yellow anions: Hydroxyl ions produced by electrolysis of water on the surface of the aluminium electrode. **c** Evolution of the current response under voltage pulses of constant polarity. Successive positive (negative) pulses result in an increase (decrease) in conductance. **d** An experiment emulating STP in fibre-based memristor. The upper plots show input voltage pulse waveforms; the bottom panel shows current response. **e** Long-term effects of the potentiation and depression on the fibre memristor's conductance. Each excitation is followed by a reading pulse. **f** The working principle of the circuit for actuation and readout of individual memristor fibres. **g** Cyclic write/erase results for 2 × 2 memristor array, only node A1 is being written/erased, while the whole array is being read. **h** Digital photograph of the 6 × 6 memristor array embedded in a cotton textile. **i** Schematic diagram of the all-textile memristor array device. **j** Results of a test for write/erase pulsing for 6 × 6 memristor array to produce patterns of stored data.

with pulsed stimulations (Fig. 4d). The applied voltage excitation was −6 V, and the duration (marked $t_p$) and interval between the excitation pulses (marked $\Delta t$) were both 50 ms. In the response current, the positive current portion (a small spike at the beginning of the impulse)

is due to the charging current response brought about by the applied voltage polarity switching. After applying multiple reverse voltage excitations, the current response dropped by approximately 0.5 mA. Inspired by long-term potentiation of synapse, we exploited the

phenomenon that the electrical conductance of metallic aluminium undergoes similar changes upon the formation or elimination of non-conductive oxidation products on its surface, enabling memristor-based computing (Fig. 4e). We access it by applying multiple "write" voltage spikes and using very small "read" pulses, which do not change memristor's state if the pulse voltage is small enough. The state of the memristor can also be reset to its original value by applying the opposite "erase" voltage spike. After 2 consecutive "long write" and 2 "long erase" cycles, the performance of the device remains stable. The working principle of the circuit is shown in Fig. 4f.

As shown in Fig. 4g, we fabricated a $2 \times 2$ memristor array with four nodes ($A_1$, $A_2$, $B_1$ and $B_2$). During the test, only the $A_1$ point is subjected to the "write" and "erase" operations. When writing to point $A_1$, the conductance of the other points was not affected significantly. After two cycles of "writing" and "erasing", the array's overall conductivity (and corresponding individual memristor states) remains stable (Supplementary Fig. 22). Furthermore, we designed a $6 \times 6$ textile memristor array on cotton as shown in Fig. 4h, i. In this architecture, we sewed aluminium yarn as warp and carbon yarn as weft on the surface of commercial cotton textile, and the hydrogel layer was deposited between the textile pores by spraying (Supplementary Fig. 23, Supplementary Movie 5 and Supplementary Note 3). The spraying process can keep the resistance between each node within a stable range (Supplementary Fig. 24). Each intersection of the two yarns forms an independent memristor node, which allows the textile to still maintain good breathability (Supplementary Fig. 25). Upon application of necessary voltage, the aluminium wire covered with gelatine will be corroded, and hydroxide will be produced on the surface, inducing a memristor-like behaviour of the junction. This vertical structural design, by controlling a reasonable write threshold voltage, can prevent write crosstalk between adjacent memristors (Supplementary Fig. 26). Based on a $6 \times 6$ memristor array, a specific pattern can be encoded by "writing" at the specific point by application of the actuating voltage; the conductance of this specific intersection will then be significantly reduced. Then, by performing a "read" operation on the entire array, it can be precisely recorded, which individual junctions were "written", and which remain in their native state. Analogously, application of an "erase" pulse allows to erase the written information, and perform the "writing" again. To demonstrate this, we have "written" and "read" the letters "B", "M", "H", and "T" onto the memristor array can be seen in the Fig. 4j. Furthermore, we developed a portable microprocessor to control the writing and reading processes of the textile memristor array (Supplementary Figs. 27, 28, Supplementary Note 4).

## Discussion

We have developed a single-fibre tunable logic and memory electronics (FLAME). FLAME is based on the corrosion-passivation effect at the electrode/electrolyte interface, eliminating the need to design complex carrier heterojunction interfaces and significantly reducing the difficulty of manufacturing one-dimensional fibre electronic devices. The electrical behaviour of the FLAME can be controlled by adjusting the pretreatment voltage and voltage application time to make it behave like a diode or a memristor, which are critical for reconfigurable textile electronic systems. The diode-like performance in FLAME can be continuously operated long-time (more than 10000 cycles) and a high voltage range of $\pm 8$ V (but not limited to), which is significantly higher than the operating voltage range of current state-of-the-art anion/cation heterojunction devices. We also demonstrate the application of FLAME in AND and OR logic gates, neuromorphic synapses, and textile memristor arrays. While the feasibility of FLAME for logic and memory applications has been demonstrated, several engineering challenges remain for practical textile integration. Future

work is required to enhance the continuous fabrication of electronic fibres and improve their environmental stability, particularly regarding hydrogel dehydration and weather resistance in complex operating conditions.

## Methods

### Ethics statement

All tests involving human participants were considered by the ETH Zurich Ethics Commission, establishing that the project is not considered human subject research and thus exempt from approval. Written and signed informed consent of all participants was obtained prior to tests and inclusion in this study.

### Materials

Aluminium yarn (FibreCoat GmbH, Germany), Hollow silicone rubber fibre (Thermo Scientific Chemicals, USA), Buffers solution with pH values of 4, 7, and 10 (Hanna Instruments, Romania), Carbon black Vulcan XC72R (Nanografi, Turkey), Conductive carbon fibre yarn (Thermo Scientific Chemicals, USA), polydimethylsiloxane (PDMS) (Dow Corning 184, US). Porcine Gelatine (G1890, Sigma-Aldrich). All materials were used as received without further purification.

### Preparation of stretchable resistive fibre

Carbon black and PDMS were mixed in a mass ratio of 1:4, fully mixed using a stirrer for 15 min, and vacuum degassed for 10 min. The mixed carbon slurry was then injected into the hollow silicone rubber fibre and cured in an oven at 80 °C for 2 h. A stretchable resistive fibre with a resistance value in the order of 500 kΩ was obtained for a textile logic gate circuit.

### Preparation of fibre-based diode

First, the conductive aluminium yarn was inserted into the hollow silicone rubber fibre, which was then stretched to 1.5 times its initial length. The stress was then released, allowing the silicone rubber to shrink, which in turn makes the internal conductive yarn recoil. Under the friction of the inner silicone wall, the aluminium yarn formed a self-organised built-in helix structure. Second, gelatine powder was dissolved in a mixture of commercial pH-buffer solution and glycerol (mass ratio 3:2) to prepare a gelatine solution with a concentration of 15 wt%, followed by stirring at 80 °C for 30 min and ultrasonic degassing. Finally, the gelatine solution was injected into the hollow centre of the silicone rubber fibre, and low-temperature cross-linking was performed in a 4 °C refrigerator to obtain the gel diode fibre. The conductive carbon fibre yarn was then used to cap the part of the fibre not touching the aluminium wire to form the second electrical terminal of the device.

### Preparation of textile-based memristor array

A sewing/embroidery machine was used to fix the conductive carbon yarn and aluminium yarn on both sides of the cotton textile to form a $6 \times 6$ textile array. Then we used a spray gun to deposit gelatine-based hydrogel on the surface of the cotton textile. The hydrogel solution diffused along the textile pores to the interface of the aluminium yarn and the carbon fibre yarn under the capillary effect, thus forming a memristor array.

### Material characterisation and device measurements

The X-ray photoelectron spectroscopy (XPS) characterisation was carried out using a PHI Genesis (Physical Electronics / ULVAC-PHI) device. The analysis setup is accredited, calibrated, and maintained according to ISO/IEC standards. The XPS data is processed by CasaXPS (V2.3.26, Casa Software Ltd., UK). The morphologies of various samples were examined using a scanning electron microscope (JEOL JSM-

7100F). The elemental distribution on the samples surface was ana-lysed by elemental mapping using a scanning electron microscope (JEOL JSM-7100F). Current-voltage (I-V) characteristics were measured by using a potentiostat (μStat-i400, MetrOhm DropSens) in a two-electrode configuration. The electrical performance of the memristor devices was characterised using a Keysight B2912B precision source/measure unit. The mechanical tests were conducted using an electric dynamic test instrument (zwickiLine Z1.0, Zwick Roell).

## Data availability
Relevant data supporting this study are available within the article and the Supplementary Information file. All the numerical data generated in this study are provided in the Supplementary Information/Source Data file in the form of an Excel file with pages corresponding to each presented graph. All data, including images, are available from the corresponding author upon request. Source data are provided with this paper.

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

## Acknowledgements

This research was funded in whole or in part by the Swiss National Sci-ence Foundation (SNSF) grant no. 219231. We acknowledge the dis-cussion with the BHMT group members. The authors gratefully acknowledge the ETH Zurich ScopeM centre for their support and assistance in this work. The authors gratefully acknowledge Dr. Roman Heuberger of the RMS Foundation company for his support and assis-tance in this work.

## Author contributions

W.Y., Y.L., and C.M. conceived the project and designed the experiments. W.Y. and Y.L. fabricated the samples and devices, ran the experiments, analyzed the data, and wrote the initial draft of the manuscript. M.R.C. designed and implemented a standalone circuit and PCB enabling programmable read/write operations on textile memristor arrays. W.Y., Y.L., and A.S. performed the initial conceptualisation and analysis on the electrochemical part. W.Y., A.S., and C.M. reviewed the manuscript. W.Y. and C.M. supervised the project.

## Competing interests
The authors declare no conflict of interest.
