## [Transparent Peer Review file · Nature Communications]

In-fibre logic and memory via tunable passivation–corrosion

Corresponding Author: Dr Weifeng Yang

Version 0:

Reviewer comments:

Reviewer #1

(Remarks to the Author)

This manuscript reports a fibre-based electronic device that integrates logic and memory functions through aluminium/electrolyte passivation-corrosion effects. The idea is interesting, but there are several important problems related to the principle and the practical application potential. Overall, I recommend rejection or at least major revision, following are some specific comments:

1. Could the fabrication of this fibre device be made continuous?
2. The rectification principle of the diode originates from the electrochemical reaction of the aluminium core. Therefore, the response time may be relatively slow. For example, in Figure 3F the rectification experiment shows obvious hysteresis. This could affect practical applications, and the authors are advised to provide an explanation.
3. Since the gel electrolyte is acidic/alkaline, will it corrode the aluminium core during long-term storage, thereby degrading device performance?
4. The authors should clarify the differences in sample size and integration methods between Figure 1B and the later figures. For example, in Figures 3G and 4H the fibres appear much thicker and are stitched onto textiles using auxiliary threads, rather than being directly woven. I would like to know whether FLAME can at this stage truly be woven into textiles as functional yarn.
5. Figure 4H and Video S4 show a memristor array fabricated on the textile surface by spraying hydrogel solution directly over a large area. This process lacks reproducibility and rigour, and it greatly limits practical applications. Moreover, this fabrication method is very different from the earlier fibre device, giving a strong sense of disconnection.

Reviewer #2

(Remarks to the Author)

This manuscript reports a reconfigurable fiber device (FLAME) that can be switched between diode-like rectification and memristive behavior by programming the Al/hydrogel interface chemistry. The FLAME is not dependent on the intricate carrier heterojunction interfaces and can be stretched by ~50% and woven into ordinary fabrics. On-fabric, it demonstrates AC rectification, OR/AND logic gates, synaptic short-term plasticity (STP), and write/read/erase operations in two-dimensional memristor arrays. The concept is original and timely for soft/wearable electronics. The results are overall convincing and well-presented. I recommend acceptance after minor revisions listed below.

1. Lines 107–111 state that “At pH 4–7, aluminium surface...and yields a forward-cutoff diode” and lines 236–240 further note that “In mildly acid (pH=4), applying a positive voltage...higher reverse current forward current”. Taken together, this seems to imply that under pH=4 operation the oxide keeps thickening, degrading the memristive response until a diode-like state emerges. Does this mean the memristor–diode transition is irreversible under acidic and neutral conditions?
2. Figure 3F presents the long-time rectification test only at pH=4. This suggests that oxide passivation at pH=4 stabilizes the diode behaviour, whereas, per the previous comment, memristive behaviour under the same acidic medium might drift as the oxide thickens—implying that pH=4 is better suited for the diode mode. Conversely, at pH=10, facile formation/dissolution of $\text{Al}(\text{OH})_3$ enables rapid “write/erase” spikes, seemingly favouring memristor operation.
3. At pH=10, the device relies on repeated formation and dissolution of $\text{Al}(\text{OH})_3$ on the aluminium yarn. While this underpins

fast write/erase, it could also lead to irreversible damage of the Al conductor (pitting, mass loss, roughening). What is the device lifetime under alkaline operation?

4. The manuscript mixes British and American conventions across spelling and style. For example, the main text alternates between “fibre” and “fiber”, and between “aluminium” and “aluminum.” It also alternates “Figure” and “Fig.” in captions/text. Please standardize language and notation per the journal’s style guide (spelling, hyphenation, units/spacing, symbols, pH formatting, figure/table references, etc.).

Reviewer #3

(Remarks to the Author)

This manuscript presents a novel single-fiber logic and memory device based on a passivation–corrosion mechanism, enabling reconfigurable logic and memory functionalities without relying on conventional rigid heterojunction structures. The device presents good flexibility, stretchability (up to 50%), and compatibility with textile weaving processes. The concept of the device is interesting. However, the following questions need to be answered:

1. It is recommended to supplement the device-to-device variation and environmental durability (such as temperature/humidity and long-term stability).
2. In high-density arrays, can logic voltages inadvertently alter the state of adjacent devices? How can interference between logic and memory be prevented?
3. In Figures 3I and 3K, the logic gate circuits (OR and AND) are demonstrated under 50% strain. The performance variation after cyclic stretching should be investigated.
4. As a typical all-textile device, its washability, breathability, and biocompatibility should be further characterized.
5. Since the textile devices are worn on the human body, how about dynamic body movements, such as twisting or pressure, affecting their performance? Moreover, beyond its electrical properties under static stretching, the performance changes of the device under dynamic strain conditions should be further investigated.
6. In Figure 4, considering the wearable merit of the memristor array, the corresponding portable control/acquisition circuit should be designed, such as the microcontroller for multi-channel signal acquisition rather than AWG and voltage source.

Version 1:

Reviewer comments:

Reviewer #1

(Remarks to the Author)

The authors have provided detailed responses to my five previous comments, which I largely find satisfactory. However, I still have one question that I would like the authors to address.

With respect to the response to Comment 5, the authors have demonstrated the scalability of spray-coating hydrogels onto textiles. However, my original concern that this approach “greatly limits practical applications” was not answered. From a wearable-textile perspective, a fabric that is covered with a wet hydrogel layer would be difficult to use in practice, as it would significantly compromise wearing comfort (see Fig. S23F). In addition, during long-term wear, dehydration of the hydrogel is likely to affect the overall device stability. If large-area encapsulation using silicone were applied, similar to the single-fibre devices, the breathability of the textile may be lost. This appears to represent a fundamental contradiction that has not yet been adequately addressed.

Reviewer #2

(Remarks to the Author)

Reviewer #3

(Remarks to the Author)

The revised manuscript has adequately addressed the previous comments, and it is now recommended for publication as is.

Response Letter to Reviewers

We are sincerely grateful to the reviewers for their time and valuable comments on our manuscript “*In-fibre logic and memory via tunable passivation–corrosion*” (NCOMMS-25-69485-T). The manuscript has been carefully revised in light of the reviewers’ comments and suggestions. All the changes in the manuscript and supporting information made in this revision are highlighted in yellow in the “Revised Manuscript” and “Revised Supporting Information”, respectively. A detailed point by point response to the comments and suggestions of the reviewers is provided below.

Reviewer #1 (Remarks to the Author):

This manuscript reports a fibre-based electronic device that integrates logic and memory functions through aluminium/electrolyte passivation–corrosion effects. The idea is interesting, but there are several important problems related to the principle and the practical application potential. Overall, I recommend rejection or at least major revision, following are some specific comments:

Response: We are grateful to the reviewer for all the valuable comments and suggestions, which have helped to improve this manuscript significantly. You will find our replies to the questions point by point below.

1. *Could the fabrication of this fibre device be made continuous?*

Response: Thank you, this is a very important question, since any fiber or textile needs to be produced in very high quantities. Continuous fabrication of electronic fibers is a fundamental prerequisite for the industrialization of smart textiles. We will explain potential solution to upscaled fabrication and potential pitfalls from the following two aspects:

(1) Potential schemes for the continuous fabrication of electronic fibers:

Drawing upon our established expertise in fiber manufacturing, we demonstrate that electronic fibers with integrated helical structures are indeed compatible with continuous production. As illustrated in Fig. R1, the proposed fabrication pipeline consists of three sequential stages: I. electrolyte coating, II. sheath encapsulation, and III. *in-situ* helix formation.

The self-organizing mechanism of the internal helix (Step 3) is based on our previous methodology ([1] *Adv. Mater.* **2021**, 33, 2104681; [2] *Nano Energy*, **2023**, 107: 108171). Specifically, thermoplastic silicone rubber (TSR) is utilized as the sheath material. It is processed via a twin-screw extruder through a high temperature melting zone to form a hollow tube. As depicted in Fig. R1a, as the sheath material reaches the extrusion die, an Al yarn is introduced co-axially to the silicone tube and co-extruded simultaneously. Upon passing compaction roller 1, the elastic sheath is subjected to a controlled longitudinal strain induced by a calibrated weight, while the core yarn is drafted in tandem. Subsequently, as the fiber passes compaction roller 2, which rotates at a lower rate, the pre-strain is released, causing the elastic sheath to recoil to its equilibrium length. Driven by the interfacial friction between the sheath and the core, the Al yarn undergoes constrained buckling to naturally form a periodic internal spiral structure. the geometric parameters and resultant maximum strain capacity of the fibers can be precisely tuned by modulating the magnitude of the applied pulling force.

(2) The reasons why fibers are not prepared continuously (Performance trade-offs) :

While technologically feasible, in practical applications, longer is not always better for such devices based on interfacial electrochemical effects. **Positive correlation between resistance and length:** we have fabricated and tested fibers ranging from 10 cm to 75 cm in length (see Figs. R1b-d). Experimental

data show that the internal resistance of the fiber increases significantly from approximately 2.8 k Ω to approximately 25.8 k Ω with increasing length. **Charge transport efficiency:** Since this device relies on carrier migration in the electrolyte, excessively long fibers increase the carrier migration path (Figure R1c). **Logic delay:** Increased resistance directly leads to increased signal transmission delay (RC delay), thus reducing the response speed of logic operations. Therefore, to balance flexibility and electrical response performance, we optimized the device length within the current range to ensure optimal dynamic performance while maintaining logic/memory functionality. While upscaled production technique for such small-segmented fibers can also be elaborated, it would require custom non-industry standard equipment. However, assembly of shorter fibre form devices from separate continuously produced fibers is also within the realm of possibilities.

Fig. R1. Continuous manufacturing potential and length-dependency analysis of FLAME fibres. (A) Proposed multi-stage continuous fabrication pipeline; (B) Photographs of FLAME fibres fabricated at different lengths (10 cm to 75 cm). (C) Schematic illustration showing that increased fibre length leads to a longer carrier migration path. (D) Measured resistance as a function of fibre length, showing an increase from ~2.8 k Ω to ~25.8 k Ω , which serves as the primary scientific

constraint for device length optimization.

2. The rectification principle of the diode originates from the electrochemical reaction of the aluminium core. Therefore, the response time may be relatively slow. For example, in Figure 3F the rectification experiment shows obvious hysteresis. This could affect practical applications, and the authors are advised to provide an explanation.

Response: We sincerely appreciate the reviewer’s insightful observation regarding the response time and hysteresis. These ion-based diodes, operation of which is based on interfacial migration/interfacial chemical reactions, indeed often exhibit longer relaxation times. The main reasons for this include: **(1) the migration rate of the ions themselves in the electrolyte;** **(2) the interfacial Faradaic reactions.** Both processes take longer than electron migration, which leads to relaxation in the rectification process. Compared to hole/electron heterojunction diodes, these ion-based diodes do not have an advantage in response time. However, they show significant advantages in biocompatibility, flexibility, stretchability, and manufacturing complexity, thus possessing potential applications in implantable electronics, electronic skin, and fabric electronics, such as low-frequency rectification and Boolean logic gates (AND Gate, OR Gate). We also provide rectification data and response times for ion-based diodes from other studies, as shown in Table R1.

Table R1 Comparison of Ionic-based diode device on hysteresis time

	Ionic current rectification	Hysteresis time
This work		~5 s
[1] Science 2024, 386, 1024–1030		~10 s
[2] Adv. Funct. Mater. 2012, 22, 625–631		~20 s
[3] Science 367, 773–776 (2020)		~10 s

● **Our revision of the manuscript:**

We added Tab. R1 as Tab. S1. Relevant modifications have been highlighted in red within the revised supplementary materials.

3. Since the gel electrolyte is acidic/alkaline, will it corrode the aluminium core during long-term storage, thereby degrading device performance?

Response: This is an interesting question since the described device operation is indeed based on ‘corrosion’ of aluminum, but storage in non-electrified state should also be considered. We conducted in-depth research on the stability of aluminum cores under acidic and alkaline conditions, focusing on both theoretical mechanisms and experimental verification.

(1) Passivation stability under acidic conditions (pH = 4): the experimental environment in the present study was between pH 4 and 10, which can be plotted on the Pourbaix potential-pH diagram for aluminum (Fig. R2A, the red dashed box). In an acidic environment (e.g., pH = 4), this potential range falls within the passivation region of aluminum. At this point, the oxide layer formed on the aluminum surface is dense, and the equilibrium concentration of Al^{3+} in the solution is extremely low ($< 10^{-6}$), meaning the dissolution rate of the oxide layer is negligible. To verify its long-term storage stability, we continuously monitored the OR logic gate fabric for 28 days (Fig. R2E). The results showed that the device maintained stable logic output for up to four weeks, without any performance degradation due to corrosion, demonstrating long-term reliability under acidic conditions.

Fig. R2 Theoretical and experimental evaluation of long-term stability. (A) Pourbaix diagram of aluminium, showing the experimental pH range (4–10) within the stable passivation and controlled corrosion regions. **(B)** Surface morphology in acidic conditions (pH = 4). **(C)** Morphological evolution and reversibility in alkaline conditions. **(D, E)** Long-term stability test of the OR logic gate over a 28-day storage period, confirming consistent voltage output.

(2) Controlled corrosion mechanism under alkaline conditions (pH = 10): Under alkaline conditions (pH = 10), the repeated formation and dissolution of $\text{Al}(\text{OH})_3$ on the aluminum surface, while fundamental for rapid read/write operations, can also cause unwanted physical damage. Our morphological evolution study revealed two distinct corrosion stages: Stage 1: Horizontal corrosion propagation stage (Figs. R2A, B). In the initial stages of the cyclic test (approximately the first 50 cycles), characteristic square corrosion pits form on the aluminum surface, and the corrosion area

gradually expands horizontally. Electrically, the corrosion current steadily increases with the number of cycles in this stage (Fig. R2E, first 50 cycles), reflecting the increase in effective reaction area. Stage 2: Vertical corrosion propagation stage (Figs. R2C, D). After more than 50 cycles, the pristine metal surface is depleted, and the majority of the corrosion now occurs in the electrode interior, which manifests in formation of more vertically oriented (i.e. normal to the surface) deep corrosion pits. The effective specific surface area remains relatively unchanged at this point, therefore, the I-V curves of the memristor remain almost identical, overlapping each other during the 50 to 250 cycle periods (Fig. R2E), with no significant performance degradation observed. Experiments demonstrate that despite surface corrosion, the FLAME device maintains stable cycling for at least 250 cycles at pH = 10. Simultaneously, the sufficient initial thickness of the aluminum electrode (~500 μm , which is orders of magnitude larger than observed pitting damages) ensures that the corrosion process does not lead to irreversible fracture, guaranteeing functional reliability under alkaline operation.

Fig. R3 Morphological evolution and electrochemical stability of the Al electrode under alkaline operation (pH=10). (A, B) SEM images and schematic illustration of the surface horizontal corrosion stage (initial 50 cycles), showing the formation and expansion of square pitting on the Al surface. (C, D) SEM images and schematic illustration of the vertical corrosion stage (50–250 cycles), where the corrosion propagates into the bulk while maintaining a stable active surface area. (E) Corresponding I-V curves of the FLAME memristor over 250 cycles.

- **Our revision of the manuscript:**

We added Fig. R2A as Fig. S1A, and Fig. R3 as Fig. S21. Relevant modifications have been made within the revised manuscript and supplementary materials (highlighted in red).

4. The authors should clarify the differences in sample size and integration methods between Figure 1B and the later figures. For example, in Figures 3G and 4H the fibres appear much thicker and are stitched onto textiles using auxiliary threads, rather than being directly woven. I would like to know whether FLAME can at this stage truly be woven into textiles as functional yarn.

Response: Thank you, indeed the fiber thickness and textile integration were not clearly presented in the previous version of the manuscript.

(1) Clarify the differences in sample size and integration methods:

- **Difference Sample Size:** indeed, the approach used in our work can be applied to make fibers with different diameters, as shown in Figure R1d, with fiber diameters ranging from 0.5mm, 0.75mm, and 1mm. However, In the manuscript, the fiber diameter in Figure 1B is 0.5mm, and the fiber diameter in Figure 1f is also 0.5mm. Figures 3G and H are 3D schematic diagrams instead of photos, so they look very coarse. Figures 3I and K are photographs of the actual objects corresponding to Figures 3G and H, with a fiber diameter of 0.5mm. In Figure 4H, the diameter of the aluminum/carbon electrode in the electronic fabric is the same as that of the electrode mentioned earlier.
- **Difference Integration Methods:** The fabric in Figure 1F is produced using a woven process; the fabric in Figures 3I and J is produced using a digital embroidery process; the fabric in Figure 4H is produced using a digital embroidery process.

(2) FLAME woven into textiles as a functional yarn :

To further demonstrate the practical viability of FLAME, we have conducted integration trials using a commercial narrow band weaving machine, as shown in Fig. R4 A–C. We also provide a video to prove the feasibility of machine weaving (Movie S1). Unlike manual stitching or simple auxiliary thread fixation, FLAME fibers were successfully treated as functional yarns and directly incorporated into the textile structure as a warp yarn in an elastic woven fabric during the automated weaving process. The resulting electronic textiles (Fig. R4E–F) exhibit high integration density and structural integrity, maintaining the intrinsic tactile properties of traditional textiles. Furthermore, as demonstrated in the tensile tests in Fig. R4G, the woven electronic textile maintains stable performance and structural consistency under large mechanical deformations, specifically within a stretchability range of 0% to 50%.

Fig. R4. Industrial-scale integration and mechanical characterization of FLAME fibers. (A–C) Digital photographs showing the integration of FLAME fibers into textiles using a commercial industrial weaving machine. (D) Photographs of FLAME fibers with various diameters (0.5 mm, 0.75 mm, and 1.0 mm). (E, F) Magnified views of the woven electronic textile, illustrating the seamless integration of FLAME as functional yarns within the fabric structure. (G) Photographs of the electronic textile under different tensile strains (0%, 25%, and 50%), showcasing its excellent structural integrity and mechanical stretchability for wearable applications.

- **Our revision of the manuscript:**

We added Fig. R4 as Fig. S6. We replaced the previous Fig. 1F in the manuscript with the industrial weaving textile. Relevant modifications within the revised manuscript and supplementary materials are highlighted.

5. Figure 4H and Video S4 show a memristor array fabricated on the textile surface by spraying hydrogel solution directly over a large area. This process lacks reproducibility and rigour, and it greatly limits practical applications.

Response: We sincerely appreciate the reviewer’s insight. We would first like to mention that in the textile industry, there are many mature precedents for integrating functional materials onto large-area fabric surfaces using spraying or automated coating processes, such as the mass production of waterproof coatings (<https://www.sono-tek.com/industry/glass-industrial/advanced-textiles/>), smart heating fabrics (<https://www.spray.com/en-eu/applications/coating>), and antibacterial functional fabrics (<https://www.fabricanltd.com/about/technology/>). This fully demonstrates the high potential of using spraying technology to prepare smart textiles in an upscaled, industry-standard way.

To further demonstrate the repeatability and rigor of this process experimentally, we fabricated a large-area fabric array containing four functional regions with a total size of approximately 40 cm × 170 cm (Fig. R5B-D). We performed multi-point independent measurements on 20 random cells spanning 170 cm. The statistical results (Fig. R5A) show that the initial resistance at all test points is consistently hovers around 40 kΩ, with minimal performance fluctuations both within and across

regions. This detailed data not only verifies the excellent repeatability of the FLAME device fabrication process in large-area integration but also proves that it meets the rigorous requirements of future large-scale production.

Figure R5. Characterization of device-to-device variation in a large-area memristor textile array. (A) Statistical box plot of the initial resistance values for 20 randomly selected memristor units across four different functional areas of the large-scale device. **(B-D)** Digital photographs of the large-scale integrated textile (approx. 40×170 cm) containing multiple memristor logic areas, showcasing the reliability of the FLAME fabrication process.

● **Our revision of the manuscript:**

We added Fig. R5 as Fig. S24. Relevant modifications have been made and are highlighted within the revised manuscript and supplementary materials.

Thank you again for your valuable comments and suggestions!

Reviewer #2 (Remarks to the Author):

This manuscript reports a reconfigurable fiber device (FLAME) that can be switched between diode-like rectification and memristive behavior by programming the Al/hydrogel interface chemistry. The FLAME is not depended on the intricate carrier heterojunction interfaces and can be stretched by ~50% and woven into ordinary fabrics. On-fabric, it demonstrates AC rectification, OR/AND logic gates, synaptic short-term plasticity (STP), and write/read/erase operations in two-dimensional memristor arrays. The concept is original and timely for soft/wearable electronics. The results are overall convincing and well-presented. I recommend acceptance after minor revisions listed below.

Response: Thank you for your assessment of our work and your comments/edits. They have helped us significantly improve the manuscript in this revision round. You will find the responses to your comments below.

1. Lines 107–111 state that “At pH 4–7, aluminium surface...and yields a forward-cutoff diode” and lines 236–240 further note that “In mildly acid (pH=4), applying a positive voltage...higher reverse current forward current”. Taken together, this seems to imply that under pH=4 operation the oxide keeps thickening, degrading the memristive response until a diode-like state emerges. Does this mean the memristor–diode transition is irreversible under acidic and neutral conditions?

Response: Thank you for this observation. We fully agree with your point: the transition from memristor to diode is indeed irreversible under acidic and neutral conditions. The core reason lies in the difference of the thermodynamic stability of the anolyte layer (electrolyte near the anode) under different pH environments:

(1) Acidic conditions (e.g., pH = 4): As shown in the Pourbaix diagram of Fig. R6A, this potential range is within the passivation region of aluminum. The oxide layer structure formed at this time is dense, and the equilibrium concentration of Al^{3+} in the solution is extremely low ($< 10^{-6}$ M), meaning that the dissolution rate of the oxide layer is almost negligible. Therefore, although we can adjust the device to exhibit memristor or diode characteristics by controlling the pre-applied voltage time, this increase in thickness is unidirectional and irreversible (as shown in Fig. R6B).

(2) Alkaline conditions: In contrast, the $\text{Al}(\text{OH})_3$ products formed under alkaline conditions are generally more porous and have lower thermodynamic stability. By applying a reverse voltage, the dissolution or stripping of the oxide layer (erasure process) can be effectively induced, thereby achieving a reversible conversion between diode and memristor characteristics (as shown in Fig. R6C).

Fig. R6 Electrochemical stability and morphological evolution of Al-based devices across different pH levels. (A) Potential–pH (Pourbaix) diagram for Aluminum. **(B)** Surface morphology in acidic conditions (pH = 4). **(C)** Morphological evolution and reversibility in alkaline conditions.

● **Our revision of the manuscript:**

We added Fig. R6A as Fig. S1A. Relevant modifications have been made and are now highlighted within the revised manuscript and supplementary materials.

2. Figure 3F presents the long-time rectification test only at pH=4. This suggests that oxide passivation at pH=4 stabilizes the diode behaviour, whereas, per the previous comment, memristive behaviour under the same acidic medium might drift as the oxide thickens—implying that pH=4 is better suited for the diode mode. Conversely, at pH=10, facile formation/dissolution of Al(OH)₃ enables rapid “write/erase” spikes, seemingly favouring memristor operation.

Response: Thank you for the constructive comment. Indeed, we fully agree with your point that the pH environment of the medium determines the optimal operating mode of the device (Diode mode vs. Memristor mode).

(1) Diode preference under acidic conditions (pH=4): Due to the dense and thermodynamically stable oxide layer produced by anodic oxidation in acidic environments (negligible dissolution rate), the oxide layer exhibits irreversible unidirectional thickening with increasing operating time or continuous voltage. While this physical process causes initial drift in the operational output of the memristor, it also provides a solid structural foundation for long-term stable diode rectification characteristics. Figure 3F shows long-term rectification testing at pH=4 to demonstrate application potential as a stable diode in this pH.

(2) Memristor preference under alkaline conditions (pH=10): In alkaline environments, the generated Al(OH)₃ products are characterized by a loose structure and exist in a dynamic equilibrium of "easy formation - easy dissolution." This instability, however, endows the device with excellent write/erase capacity. The oxide layer can be thinned or removed by application of reverse bias, thus avoiding unidirectional drift of the operational characteristics. Therefore, an alkaline environment is indeed an ideal medium for achieving high-performance, high-cycle-frequency memristor behavior.

Our research reveals a new strategy for "customizing" device function through ambient pH: acidic environments solidify diode characteristics, while alkaline environments activate memristor cycling. We have supplemented the discussion section of the paper with relevant arguments based on your suggestions to clarify the optimal application scenarios under different pH conditions.

- **Our revision of the manuscript:**

We added a discussion of preferences for diode-like and memristor-like behaviour in the developed device under acidic/alkaline conditions as a supplement to Note S2.

3. At pH=10, the device relies on repeated formation and dissolution of $Al(OH)_3$ on the aluminium yarn. While this underpins fast write/erase, it could also lead to irreversible damage of the Al conductor (pitting, mass loss, roughening). What is the device lifetime under alkaline operation?

Response: This is indeed an important question from the standpoint of practical applications. We fully agree that the repeated formation and dissolution of $Al(OH)_3$ on the aluminum electrode surface under alkaline conditions (pH=10) is the physical basis for achieving rapid read/write operations, but it may also cause physical damage to the aluminum conductor.

To assess device lifetime, we have conducted a detailed study on the morphological evolution and electrical stability of the aluminum electrode surface during cycling (see Fig. R7): Two stages of the corrosion mechanism: **Stage 1:** Horizontal corrosion stage (Fig. R7A, B). In the initial stage of cycling (approximately the first 50 cycles), characteristic square corrosion pits gradually form on the aluminum electrode surface. As the number of cycles increases, the corrosion area gradually expands horizontally. In terms of electrical performance, this stage corresponds to the I-V curve in Figure R6E for the first 50 cycles; the corrosion current steadily increases with the number of cycles, reflecting the increase in effective reaction area. **Stage 2:** Vertical corrosion stage (Figs. R7C, D). After more than 50 cycles, the entire aluminum electrode surface is covered by corrosion, at which point the corrosion shifts to a vertical penetration into the electrode interior. During this stage, the effective specific surface area of the electrodes remained relatively constant overall. Therefore, the I-V curves of the memristor exhibited extremely high overlap during the 50 to 250 cycle periods (Fig. R7E), with no significant current performance degradation.

Device Lifetime and Reliability: Experimental results show that despite mass loss and surface roughening, the FLAME device maintained stable cycling for at least 250 cycles at pH=10 due to the electrochemical stability of the electrode surface during the longitudinal corrosion stage. Furthermore, because the aluminum electrode had sufficient initial thickness, this controlled corrosion process did not lead to irreversible conductor breakage during the experimental observation period, thus ensuring the functional reliability of the device under alkaline operation.

Fig. R7 Morphological evolution and electrochemical stability of the Al electrode under alkaline operation (pH=10). (A, B) SEM images and schematic illustration of the surface horizontal corrosion stage (initial 50 cycles), showing the formation and expansion of square pitting on the Al surface. (C, D) SEM images and schematic illustration of the vertical corrosion stage (50–250 cycles), where the corrosion propagates into the bulk while maintaining a stable active surface area. (E) Corresponding I-V curves of the FLAME memristor over 250 cycles.

- **Our revision of the manuscript:**

We added Fig. R7 as Fig. S21. Relevant modifications have been highlighted in red within the revised manuscript and supplementary materials.

4. The manuscript mixes British and American conventions across spelling and style. For example, the main text alternates between “fibre” and “fiber”, and between “aluminium” and “aluminum.” It also alternates “Figure” and “Fig.” in captions/text. Please standardize language and notation per the journal’s style guide (spelling, hyphenation, units/spacing, symbols, pH formatting, figure/table references, etc.).

Response: Thank you for the kind reminder. Based on your suggestion, we have thoroughly proofread the entire manuscript and standardized the spelling according to the following criteria:

(1) Spelling Consistency: We have standardized the spelling style throughout the manuscript to British English (e.g., using "fibre" instead of "fiber", "aluminium" instead of "aluminum").

(2) Standardization of Notation and Abbreviations: The formatting of figures and tables in the text has been standardized (e.g., the use of "Figure" and "Fig." is consistent in both the text and figure captions). The use of spaces and hyphens between units and values, as well as the formatting of symbols and pH values, has been unified.

- **Our revision of the manuscript:**

All places involving spelling corrections, symbol standardization, and formatting adjustments have been highlighted in red in the revised manuscript.

Thank you again for your valuable comments and suggestions!

Reviewer #3 (Remarks to the Author):

This manuscript presents a novel single-fiber logic and memory device based on a passivation–corrosion mechanism, enabling reconfigurable logic and memory functionalities without relying on conventional rigid heterojunction structures. The device presents good flexibility, stretchability (up to 50%), and compatibility with textile weaving processes. The concept of the device is interesting. However, the following questions need to be answered.

Response: We are thankful for the consideration of our work and all the helpful comments. You will find our replies to each of them below.

1. It is recommended to supplement the device-to-device variation and environmental durability (such as temperature/humidity and long-term stability).

Response: We thank the reviewer for their suggestion. Indeed, such analysis was lacking in the initial manuscript.

(1) Device-to-device variation : In response to reviewers' suggestions regarding device consistency, we designed and fabricated a large-area textile memristor array containing four functional regions, with a total size of approximately 40 cm × 170 cm (see Figs. R8B-D). Through multiple independent and repeated measurements of five randomly selected memristor units in each region, the statistical results (Fig. R8A) show that the initial resistance at all test points remained stable at approximately 40 kΩ, with minimal performance fluctuations both within and across regions. This data fully demonstrates the high reliability and excellent inter-device consistency of the FLAME device fabrication process during large-area integration, ensuring the functional stability of large-scale smart textiles in practical applications.

(2) Temperature stability: We further evaluated the stability of the logic gate textile under different ambient temperatures through human wear experiments. As shown in Figs. R9 A and B, we integrated the FLAME-based OR logic gate into the wristband and used a heat gun to simulate changes in skin surface temperature. The experimental results show that the device always maintains a stable high-level logic output when the input is (1, 1). As the skin surface temperature increased from 32.5 °C to 60.1 °C (Fig. R9 A), the output voltage gradually increased from about 1.8 V and stabilized at around 2.4 V (Fig. R9 A). Although the output level showed a slight upward trend with increasing temperature, it was always well above the threshold for logic judgment, demonstrating the functional reliability of FLAME under normal physiological fluctuations and extreme thermal environments.

(3) Humidity stability: We further evaluated the stability of the FLAME device under varying humidity conditions. As shown in Fig. R9 D, we placed the OR logic gate integrated on the wristband in a closed microenvironment and precisely regulated the local humidity using a humidifier. During the experiment, the relative humidity of the microenvironment, monitored in real time by a humidity sensor, increased from 35.42% to 78.43% (Fig. R9 C). During this period, the output potential of the logic gate with input (1, 1) remained consistently high and was almost unaffected by humidity fluctuations (Fig. R9 D). This excellent humidity stability is mainly attributed to the structural design of the FLAME: the electrode-electrolyte interface and electrolyte material are completely encapsulated inside a hydrophobic silicone rubber tube, maximizing the isolation of environmental moisture from the internal electrochemical processes. The experimental results demonstrate that the FLAME maintains reliable logic operation functionality even in high humidity or when the wearer is sweating.

(4) Long-term stability: As shown in Fig. R9 E, we placed the OR logic gate under normal environmental conditions and continuously monitored its logic output performance in the initial state, on day 7, day 14, and day 28. The experimental results show that during the four-week test period, when the input was (1, 1), the logic gate maintained a stable high-level output potential (approximately 2.0 V) without significant performance degradation nor functional failure. This indicates that the electrolyte system and electrode interface within the FLAME fiber have good chemical stability, and the encapsulation structure effectively prevents the loss of inner components or environmental degradation, demonstrating the reliability of the system in long-term wear and practical applications.

Fig. R8. Characterization of device-to-device variation in a large-area memristor textile array. (A) Statistical box plot of the initial resistance values for 20 randomly selected memristor units across four different functional areas. (B-D) Digital photographs of the large-scale integrated textile (approx. 40 cm * 170 cm) containing multiple memristor logic areas, showcasing the reliability of the FLAME fabrication process.

Fig. R9 Environmental durability and long-term stability characterization of the FLAME device. (A) Infrared thermography of the wearable logic gate wristband under heating, showing the skin surface temperature increasing from 32.5 °C to 60.1 °C. (B) Output voltage of the OR logic gate during the heating process; the high-level state remains stable despite a slight voltage drift. (C) Photographs of the device under varying micro-environmental humidity levels (from 35.42% RH to 78.43% RH) controlled by a humidifier and monitored by a sensor. (D) Corresponding output voltage of the OR logic gate at (1, 1) input, demonstrating excellent humidity resistance due to the protective silicone encapsulation. (E) Long-term stability test of the device over a 28-day storage period (Initial, 7 days, 14 days, and 28 days), confirming the structural and functional reliability.

● **Our revision of the manuscript:**

We added Fig. R8 as Fig. S24, and Fig. R9 as Fig. 17. Relevant modifications have been highlighted within the revised manuscript and supplementary materials.

2. In high-density arrays, can logic voltages inadvertently alter the state of adjacent devices? How can interference between logic and memory be prevented?

Response: Thank you for your question, indeed cross-talk and cross influence are important to consider in any electronic arrays. In high-density crossbar arrays, preventing crosstalk between adjacent devices due to "sneak circuit paths" is crucial. We address this issue through voltage threshold control, as detailed below:

(1) Equivalent circuit analysis: When we address a specific device (e.g., M₃₁) in the array and apply a write voltage V_0 , current will flow through adjacent devices if other lines are floating. As shown in Fig. R10 A, the nearest interference path consists of three memristors connected in series (M₃₃, M₂₃, and M₂₂). Assuming that the resistances R of the devices are similar, according to the voltage division equation, each adjacent device on the path shares only $1/3 V_0$. Write threshold control (see Fig. R10 B): The reliability of state switching depends on the intrinsic electrical characteristics of the devices. Experimental data show that the "writable voltage" (i.e., the threshold voltage for state switching) of our memristors is stable above 1 V.

(2) Anti-interference mechanism: By setting the operating voltage V_0 within the range of 1 V to 3V, precise addressing can be achieved: the target device (M₃₁) withstands the full voltage V_0 (>1 V) and can perform write or logic operations normally. Adjacent devices (M₃₃, M₂₃, M₂₂ only withstand about $1/3 V_0$ (<1 V), which is far below the device's toggle threshold.

In summary, this voltage distribution mechanism ensures that logic operations only act on the target device, effectively preventing interference to adjacent memory cells and achieving decoupling of logic and memory functions.

Taking this effect into consideration, larger arrays would be even less prone to cross-talk, as more and more elements would be on the series-connected branch of the equivalent circuit.

Figure R10. Analysis of crosstalk suppression in the memristor array. (A) Schematic of the crossbar array showing the addressing of device M₃₁ and the corresponding equivalent circuit for the sneak path involving adjacent devices M₃₃, M₂₃ and M₂₂. **(B)** Typical I-V characteristic of the memristor in Fig. 2, highlighting the "writable voltage" region (red shaded area, >1 V).

● **Our revision of the manuscript:**

We added Fig. R10 A as Fig. S26. We also added above discussion as Note S4. Relevant modifications have been highlighted within the revised manuscript and supplementary materials.

3. In Figures 3I and 3K, the logic gate circuits (OR and AND) are demonstrated under 50% strain. The performance variation after cyclic stretching should be investigated.

Response: We thank the reviewer for this comment. To ensure clarity for all readers, we have further clarified the section on operation under static strain in the revised figure caption and the main text. As for the performance variation after cyclic stretching, we have performed experiments with application of cyclic strain (50%) and the functional characteristics of the device remain virtually unchanged.

Fig. R11 OR logic gate textile under stable output potential of (1, 1) logic under 2000 cycles of strain.

4. As a typical all-textile device, its washability, breathability, and biocompatibility should be further characterized.

Response: Thank you for the reviewers' suggestions.

(1) **Washability:** To verify the durability of FLAME as an all-fabric device, we directly immersed it in 1 liter of deionized water containing 3 mL of laundry detergent, and simulated a real washing environment at room temperature with magnetic stirring at 600 rpm, without any encapsulation protection (Fig. R12A). After 1, 5, and 10 washing cycles (each cycle lasting 30 minutes followed by air drying), we tested the electrical output of its OR logic gate. The results showed that even with a high input level (1, 1), the device could still stably output a consistent high potential without significant fluctuations, fully demonstrating its excellent structural stability and chemical durability (Figs. R12B and C).

Fig. R12 Characterization of the washability and electrical stability of the FLAME textile. (A) Schematic and digital photographs of the simulated washing process. **(B, C)** Output voltage signals of the FLAME-based OR logic gate after 0, 1, 5, and 10 washing cycles.

(2) Breathability: We strongly recognize the importance of breathability for textile-based electronic devices and have conducted both qualitative and quantitative characterization studies. First, qualitative testing showed that gas molecules can smoothly pass through the submerged textile and generate continuous bubbles, directly demonstrating the openness of its porous structure. Subsequently, referring to GB/T 24218.15-2018 standard, we performed quantitative measurements on the textile before and after integration at a pressure of 100 Pa. The experimental results show that the air permeability of the logic gate fabric substrate and the integrated fiber electronic fabric are 144 mm/s and 121 mm/s, respectively (Fig. R13). The air permeability only slightly decreased after integrating FLAME fibers and remained at a high level. This indicates that our integration process effectively preserves the original breathability advantages of the textile, meeting the human comfort requirements of wearable applications.

Fig. R13 Characterization of the breathability of the FLAME device. (A) Digital photograph of the qualitative breathability test, where air bubbles successfully pass through the submerged electronic textile. (B) Experimental setup of the air permeability tester used for quantitative measurement. (C) Comparison of air permeability between the original textile substrates and the integrated electronic textiles at a pressure of 100 Pa.

(3) **Biocompatibility:** we demonstrated the safety of the developed textile devices through human skin patching experiments and structural design analysis: First, the electrolytes of the functional fibers are completely encapsulated inside the flexible fiber tubes, avoiding direct contact with the skin and physically preventing chemical irritation; second, the fabric maintains a breathability of up to 121 mm/s (see Fig. R13C), ensuring normal heat and sweat dissipation and avoiding stiffness and discomfort; finally, by photographing and observing the OR logic gate fabric worn on the wrist (Fig. R14), the results showed that no redness, swelling, itching, or inflammatory reaction occurred on the skin surface after 2 hours and 6 hours of continuous wear, fully demonstrating the good biocompatibility of this all-fabric device, making it suitable for long-term skin contact wear.

Fig. R14 Evaluation of the biocompatibility of the FLAME device. (A) Photograph of FLAME-based OR logic gate integrated into a wearable wristband. (B, C) Skin condition at the contact area after continuous wearing for 2 hours and 6 hours, respectively.

● **Our revision of the manuscript:**

We added Fig. R12 as Fig. S18, and Fig. R13 as Fig. S25. Relevant modifications have been highlighted in red within the revised manuscript and supplementary materials.

5. *Since the textile devices are worn on the human body, how about dynamic body movements, such as twisting or pressure, affecting their performance? Moreover, beyond its electrical properties under static stretching, the performance changes of the device under dynamic strain conditions should be further investigated.*

Response: Thank you for the suggestions. Considering the practical application scenarios of all-fabric devices worn on the human body, we further evaluated the electrical performance of the FLAME logic gate under dynamic motion (such as dynamic twisting, pressing, and stretching). As shown in Fig. R14, under the (1, 1) input state, the device generally maintains stable logic operation function even during continuous dynamic deformation. In the experiment, we observed that the output voltage fluctuated with the motion state. The main reason is that when the FLAME fiber is twisted, squeezed, or stretched, the geometry of the internal electrolyte channels changes, which may lead to an increase in the local resistance of the electrolyte, thus causing a corresponding fluctuation in the output voltage. However, the data shows that this voltage fluctuation range always remains within the high-level effective range of the logic judgment (approximately 1.5 V to 2.8 V), and no logic level misjudgment occurred. This proves that the FLAME logic gate has good functional robustness in complex dynamic mechanical environments and can meet the basic logic signal processing requirements in human activity monitoring.

Fig. R15 Performance characterization of the FLAME-based logic gate under dynamic mechanical deformations. Real-time output voltage curves of the OR logic gate under (1, 1) input during dynamic twisting (top), stretching (middle), and pressing (bottom).

6. In Figure 4, considering the wearable merit of the memristor array, the corresponding portable control/acquisition circuit should be designed, such as the microcontroller for multi-channel signal acquisition rather than AWG and voltage source.

Response: We agree with the reviewer that a portable control and acquisition system is essential for wearable memristor arrays. Accordingly, we designed a custom integrated control board to replace bulky lab equipment (AWG/Voltage Source). The key features are as follows:

(1) System Architecture (Fig. R16): We developed an 8×8 switch matrix controlled by a microcontroller (MCU) via cascaded shift registers. This setup enables precise addressing of any individual memristor at the (T_j , B_i) intersection while keeping unselected lines in a high-impedance state to minimize crosstalk.

(2) Integrated Operations: Write Mode: The system routes regulated programming voltages (5V, 10V, or 12V) to the selected column while clamping the selected row to GND. **Read Mode:** To avoid negative supply rails, the polarity is reversed. A read bias (0.1V, 0.5V, or 1.0V) is applied to the row, and the resulting current is routed from the column to a sensing network.

(3) Signal Acquisition and Processing: The sensing circuit utilizes a precision shunt resistor and a differential amplifier ($G=100$). Signals are filtered and digitized by the MCU's onboard ADC. To ensure accuracy, each data point is an average of 20 ADC samples, effectively reducing random noise.

(4) Portability and Validation: The entire system operates from a single 14 V supply, with all bias levels generated internally via regulators and buffers. The accuracy of the readout chain was validated using a programmable decade box, ensuring reliable resistance characterization in a wearable form factor.

We also provide the circuit diagram (Fig. R17) for the microcontroller system.

Fig. R16. Portable control and signal acquisition system for the memristor array. (A-C) PCB layout and photograph of the custom-designed memristor matrix control board. (D) Detailed circuit schematics, including the 8×8 switch matrix architecture, the current measurement circuit (amplification and filtering), and the multi-level voltage source circuit.

Fig. R17. Circuit diagram for 8x8 memristor electronic system

● **Our revision to the manuscript:**

We added Fig. R16 as Fig. S27, and Fig. R17 as Fig. S28. We also added above discussion as Note S5. Relevant modifications have been highlighted in red within the revised manuscript and supplementary materials.

Thank you again for your valuable comments and suggestions!

Response Letter to Reviewers

We are sincerely grateful to the editor and the reviewers for their time and valuable comments on our manuscript “*In-fibre logic and memory via tunable passivation–corrosion*” (NCOMMS-25-69485A). The manuscript has been carefully revised in light of the reviewers’ comments and suggestions. All the changes in the manuscript and supporting information made in this revision are highlighted in red in the “Revised Manuscript” and “Revised Supporting Information”, respectively. A detailed point by point response to the comments and suggestions of the reviewers is provided below.

Reviewer #1 (Remarks to the Author):

The authors have provided detailed responses to my five previous comments, which I largely find satisfactory. However, I still have one question that I would like the authors to address. With respect to the response to Comment 5, the authors have demonstrated the scalability of spray-coating hydrogels onto textiles. However, my original concern that this approach “greatly limits practical applications” was not answered. From a wearable-textile perspective, a fabric that is covered with a wet hydrogel layer would be difficult to use in practice, as it would significantly compromise wearing comfort (see Fig. S23F). In addition, during long-term wear, dehydration of the hydrogel is likely to affect the overall device stability. If large-area encapsulation using silicone were applied, similar to the single-fibre devices, the breathability of the textile may be lost. This appears to represent a fundamental contradiction that has not yet been adequately addressed.

Previously Comment 5: *Figure 4H and Video S4 show a memristor array fabricated on the textile surface by spraying hydrogel solution directly over a large area. This process lacks reproducibility and rigour, and it greatly limits practical applications.*

Response: We sincerely thank the reviewer for this constructive comment. We agree that the trade-off between moisture retention (stability) and air permeability (comfort) is a common challenge in wearable electronics. We would like to address this concern from the following three aspects:

- 1. Localized Integration vs. Full-garment Coating:** In many practical wearable applications, these memristor arrays are envisioned as localized “computational nodes” or “functional patches” rather than a continuous coating over the entire garment. For instance, integrating the array into specific areas like sleeves or pocket regions—which account for only a small fraction of the total garment surface—allows for robust silicone encapsulation to ensure device stability. Given the high baseline breathability of our textile (121 mm/s), the localized loss of air permeability in these small functional zones will not significantly compromise the overall thermal comfort of the wearer (Figure R1).

Figure R1 Schematic illustration of localized integration on breathable textiles. The memristor array is integrated as a functional "patch" within the garment. The overall of the textile substrate maintains its high **air permeability**, ensuring the overall wearing comfort and breathability of the garment.

2. **Potential Engineering Solutions:** To further mitigate hydrogel dehydration, several strategies can be explored in future optimizations. These include: (i) introducing humectants like glycerol to form organohydrogels that resist evaporation, and (ii) utilizing breathable yet waterproof membranes (e.g., ePTFE) to balance moisture retention and air permeability. Furthermore, industrial standard technologies such as lamination are already widely used to integrate and protect fragile components within textiles.

We would like to emphasize that the primary contribution of this work lies in the **scientific demonstration of a new scheme for constructing logic/memory devices by leveraging the tunable passivation-corrosion effects at the fibre electrode-electrolyte interface**. We have validated the feasibility of this mechanism from a fundamental and theoretical perspective. While we have demonstrated its potential in wearable forms, we acknowledge that full-scale commercialization will require systematic engineering regarding advanced encapsulation and long-term durability.

In the revised manuscript (Discussion section), we have added a dedicated paragraph addressing these trade-offs and future engineering directions: **"While the feasibility of FLAME for logic and memory applications has been demonstrated, several engineering challenges remain for practical textile integration. Future work is required to enhance the continuous fabrication of electronic fibres and improve their environmental stability, particularly regarding hydrogel dehydration and weather resistance in complex operating conditions."** This revision are highlighted in red in the "Revised Manuscript"

Thank you again for your valuable comments and suggestions!